# Allelopathic and Herbicidal Effects of Crude Extract from *Chromolaena odorata* (L.) R.M.King and H.Rob. on *Echinochloa crus-galli* and *Amaranthus viridis*

**DOI:** 10.3390/plants10081609

**Published:** 2021-08-05

**Authors:** Thanatsan Poonpaiboonpipat, Ramida Krumsri, Hisashi Kato-Noguchi

**Affiliations:** 1Department of Agricultural Science, Faculty of Natural Resources and Environment, Naresuan University, Phitsanulok 65000, Thailand; 2Department of Applied Biological Science, Faculty of Agriculture, Kagawa University, Miki 761-0795, Kagawa, Japan; ramidakrumsri@gmail.com (R.K.); kato.hisashi@kagawa-u.ac.jp (H.K.-N.)

**Keywords:** allelopathy, crude extract, Siam weed, natural herbicide, weed control

## Abstract

The present study was aimed at investigating the allelopathic effects of a crude extract from *Chromolaena odorata* (L.) R.M.King and H.Rob. (Siam weed). The effects of 70% crude ethanol extract from the whole plant, leaf, stem, and root on the germination and growth of *Echinochloa crus-galli* and *Amaranthus viridis* seedlings were evaluated using Petri-dish tests under laboratory conditions. Crude extracts from the leaf showed the highest inhibitory activity. The leaf extract (OR) was further separated by sequential solvent extraction to provide hexane (HX), ethyl acetate (ET), and butanol (BU) fractions, which were also evaluated using Petri-dish tests. The hexane fraction was significantly the most active; therefore, it was selected for formulation in a concentrated suspension and tested for its herbicidal characteristics. The formulation showed greater early post-emergence than post- and pre-emergence activities, respectively. The physiological mechanism of the formulation was tested against *E. crus-galli* and showed that chlorophyll a and b and the carotenoid contents of the leaf dramatically decreased when the concentration was increased, suggesting its ability to disrupt the process of photosynthesis. As thiobarbituric acid reactive substances also occurred in the leaf of *E. crus-galli*, this suggests lipid peroxidation and cell disruption. These results represent the possibility that *C. odorata* extract contains inhibitory compounds with herbicidal activity and could be used as an early post-emergence herbicide for weed control.

## 1. Introduction

Many plant species, including crop plants, can produce and release biologically active compounds, termed allelopathic compounds, allelochemicals, or allelochemics [1], which affect the growth and development of surrounding organisms. Allelochemicals have been proven to affect other plant species through different pathways, such as disturbing mitotic activity in the roots or seedling growth; the ion uptake rate; photosynthesis; respiration; protein formation; the permeability of cell membranes; and changing or inactivating the functions of certain hormones and enzymes [2,3,4]. Various allelochemicals can be stimulatory or inhibitory, depending on the type of active compounds or the static and dynamic availability and the target species [5]. Based on the allelochemicals’ potential, the inhibitory effects can be used as a starting point for the development of selective herbicides [6]. The use of allelochemicals could enhance sustainability in agriculture since it could reduce or replace the use of synthetic pesticides for pest management, thus causing less pollution while being safer for humans, animals, and agricultural products [7,8,9]. To date, numerous allelopathic compounds have been successfully identified in and isolated from plants or plant extracts worldwide. Benzoxazoline (BOA) exuded from the roots of several grass species; sarmentine extracted from the fruits of the long pepper (*Piper longum* L.); fatty acids, such as pelargonic acid and decenoic acid, found in many plant species; and citral, a diterpene component of many essential oils, are examples of natural products that have the potential for use as natural herbicides [10]. However, because they are expensive to synthesize, the sourcing may be limited; patent protection is an additional limitation on the use of natural products [11].

Interestingly, crude extract from allelopathic plants acted in the same way as the joint action of allelochemical mixtures, which have shown a strong inhibitory effect that might have the potential for use as a natural product [12]. Heisey [13] reported the effect of crude methanolic extract from the bark of *Ailanthus altissima* (family: Simaroubaceae), in which ailanthone is a major active compound, and which exhibited both pre- and post-emergence activity against *Amaranthus retroflexus* L., *Lepidium sativum* L., *Abutilon theophrasti* Medik., *Hordeum jubatum*, *Echinochloa crus-galli*, and *Zea mays* L. However, the crude extracts contained both active and inactive compounds. Therefore, a technique for eliminating the inactive compounds in the crude extract was used. The separation of solvent partition is a technique extensively used to separate and purify natural products [14]. Wichittrakarn et al. [12] reported that the ethyl acetate fraction from crude ethanolic extract of *Tagetes erecta* showed greater inhibition of germination and seedling growth in *Echinochloa crus-galli* than the original crude. However, one challenge in applying crude extracts for weed control is their instability and insolubility in water, making them difficult to apply on weeds. In general, crude extracts from plants contain multiple compounds, such as essential oils, that dissolve better in other organic solvents than in water. Thus, the crude extract is a pathway of interest toward producing a natural herbicide formulation. The crude extracts from plants that are insoluble in water can be combined with several carriers in order to increase or extend their efficacy, and the proper formulation might markedly affect the phytotoxic effect of a compound [15].

*Chromolaena odorata* (L.) R.M.King and H.Rob. (Siam weed) is a short-day flowering perennial shrub that flowers in winter at the start of the dry season [16]. It is an invasive weed where it has been introduced into the tropical regions of Asia, Africa, and the Pacific [17]. An extract of *C. odorata* had been reported to have several biological properties, such as anti-inflammatory effects in carrageenan-induced inflammation in rats, as well as antipyretic activity in mice [18]. The extracts have also been reported to inhibit the growth of *Bacillus subtilis*, and volatile oils obtained from the extract showed insecticidal activity against the maize grain weevil [19,20]. Ambika [21] reported that the freshly decomposed leaves of *C. odorata* suppressed crop growth. The inhibition lasted for 60 days, and the fully decomposed leaves then promoted other crop growth. A methanolic extract of *C. odorata* leaves has also been reported to inhibit seed germination and growth of *Vigna radiata* seedlings [22]. In addition, leaf extract from *C. odorata* also inhibited the weeds *Eleusine indica*, *Cyperus iria*, and *Ageratum conyzoides* [23]. Although the allelopathic properties of *C. odorata* on plant species have been published, none concern the application of *C. odorata* extracts for weed control until now.

Thus, the aims of this study were to (i) compare the allelopathic effects of extracts derived from the leaf, stem, root, and whole plant; (ii) partially isolate the active fraction from the extract and to test its activity; (iii) evaluate the herbicidal characteristic as pre-, early post-, and post-emergence of the selected crude fraction under glasshouse conditions; and (iv) explore the metabolic processes involved in producing chlorophylls and carotenoid content, as well as lipid peroxidation against *E. crus-galli*.

## 2. Results

### 2.1. Effects of Extracts from Different Parts of Chromolaena odorata on the Test Plants

The 70% ethanol extracts of the leaf, stem, root, and whole plant showed different levels of phytotoxicity towards the germination and seedling growth of *Echinochloa crus-galli* and *Amaranthus viridis* (Table 1). The leaf extracts showed the highest inhibitory effect on both species, followed by extracts from the whole plant, the root, and the stem. The leaf extracts inhibited *E. crus-galli* germination by 90% and 92.5% at 4000 and 8000 ppm, respectively, whereas other extracts and concentrations did not show a significant effect. *Amaranthus viridis* seemed to be more susceptible to *C. odorata* extracts than *E. crus-galli*. For example, the leaf extract completely inhibited *A. viridis* germination at a concentration of 2000 ppm, while this concentration did not inhibit *E. crus-galli* germination.

### 2.2. Effects of Sequential Solvent Extraction of Active Compounds on the Test Plant

The leaf extract of *C. odorata* showed the highest inhibitory effect on *E. crus-galli* and *A. viridis*, and consequently, the leaf extract was separated into different fractions by continuous partitioning with different polarities of organic solvent. The effects of the HX, ET, and BU fractions on the germination and seedling growth of *E. crus-galli* and *A. viridis* were compared with the effects of the OR fractions. The phytotoxicity of all fractions depended on the test plant species, the fraction, and the concentration (Figure 1). The HX fraction inhibited germination and the shoot and root growth of both species to a greater extent than the ET, BU, and OR fractions. The HX fraction inhibited seed germination of *E. crus-galli* by 70% and 98.75% at 2000 and 4000 ppm, respectively; both concentrations completely inhibited germination of *A. viridis*. All fractions showed a greater inhibitory effect when the concentration of the extract was increased. Active constituents of *C. odorata* are thus present in all fractions but are present at higher levels in the HX extract.

### 2.3. Pre- and Post-Emergence Herbicidal Activity

Since the HX had shown the highest phytotoxic effect in preliminary tests, it was decided to evaluate these fractions as pre-, early post-, and post-emergence herbicides. Because of the difficulty involved in dissolving the crude material, the fractions were formulated to suspension concentrate (SC) named as CH product. The delay of the emergence of both species after CH application was shown at rates of 10 and 20 kg ai/ha (Figure 2). CH product decreased the emergence of *A. vidiris* to a greater extent than *E. crus-galli*. The final emergence at 14 days after application showed that 20 kg ai/ha only reduced the percentage of the emergence of *A. viridis* and *E. crus-galli* by 25% and 12.1%, respectively. The emergence of dry shoot weight is shown in Figure 3, indicating no difference on *A. viridis*; 20 kg ai/ha reduced only the dry weight.

The CH product was spray-applied to test for efficacy against early post-emergence (7-day-old plants) and post-emergence (25-day-old) in both species. Figure 4 shows the weed control efficacy of CH at increasing times after application. The weed control efficacy increased with increasing times and rates of application. The CH product showed greater efficacy on early post- than on post-emergence application. The 10 and 20 kg ai/ha completely killed the seedlings of *E. crus-galli* in early post-emergence, while *A. viridis* was only killed at 20 kg ai/ha. The dried weight of survival seedlings shown in Figure 5 shows that over 5 kg ai/ha significantly reduced the dried weight of both species when applied as early post-emergence, while 2.5 kg ai/ha was not different from the untreated. For post-emergence application, the 20 kg ai/ha significantly reduced the dried weight of both species.

### 2.4. Chlorophyll and Carotenoid Contents

Since the CH product appears to have early-post emergence herbicide characteristics, some physiological mechanisms were investigated in *E. crus-galli*, a model species, because it is more sensitive than *A. viridis*. A decrease in the content of chlorophyll a, chlorophyll b, and carotenoid was indicated with an increase of the times after application (Figure 6A,B). The pigment content in treated leaves was not significant at 6 h after application when compared with the untreated leaves. The pigment content was initially reduced at 12 h after application. At 48 h after application, the chlorophyll a content in the untreated leaves was 4.6 mg/g DW, while the treated leaf was 2.05 mg/g DW. Likewise, the chlorophyll b content was 1.65 mg/g DW in the untreated and 0.65 mg/g DW in the treated leaves. In addition, the carotenoid content was 0.85 mg/g DW in the untreated and 0.36 mg/g DW in the treated leaves at 48 h after application (Figure 6C).

### 2.5. Lipid Peroxidation

The results of thiobarbituric acid reactive substances (TBARS) content in the sprayed leaves of barnyard grass are shown in Figure 7. The amount of TBARS increased in correlation with an increase in times. The TBARS content in untreated leaves was 3.1, 3.5, 4.5, and 5.5 nmol/g FW at 6, 12, 24, and 48 h, while the content in treated leaves was 6.5, 9.16, 15.6, and 25 nmol/g FW, respectively.

## 3. Discussion

*Chromolaena odorata* extracts affected the germination and seedling growth of *E. crus-galli* and *A. viridis*, both through inhibition and stimulation depending on several factors (Table 1). In our experiment, the observed difference between test plants found that the seed size of *A. viridis* was smaller than that of *E. crus-galli*. Therefore, one reason for the discrepancy in the inhibitory activity may be related to the difference in seed size of the test plants. This effect was also reported for *A. odorata* extracts, which were found to be more inhibitory against *E. crus-galli* than *Phaseolus lathyroides* [6]. Comparing IC_50_ values, the results showed that the *C. odorata* leaf extract had a greater phytotoxic effect on germination and on the shoot and root lengths of both *E. crus-galli* and *A. viridis* than extracts of other parts of *C. odorata* (Table 1). The effectiveness of the leaf extracts on seed germination and seedling growth was greater than the extracts from other parts, which corresponds to the reported findings of many researchers [24,25]. Leaf extracts may be more phytotoxic because the leaves are a major site of plant metabolism, and levels of secondary metabolites are higher than in other parts of the plant [26,27]. Therefore, further actions were performed with the leaf extract to separate the active fraction using sequential extraction of solvents with increasing polarity.

Our results showed that the HX fraction was the most inhibitory against tested weed species compared with other fractions (Figure 1). This may demonstrate that the most active compound is contained in this fraction. Hexane is a non-polar solvent used for extracting cooking oil from canola, sunflower, and soybean [28], suggesting that the major active compounds of *C. odorata* may be the oils or essential oils. Shentode et al. [29] reported that the hexane fraction separated from a crude methanol extract derived from *C. odorata* leaf was found to contain eugenol, copaene, caryophyllene, (−)-spathulenol, caryphyllene oxide, globulol, α-cadinal,1-hexadecanol, hexadecane, tridecane, tetracosane, pentadecanal, octadecanal, tetracosanol, and eicosane, while the volatile oil from *C. odorata* was found to contain *trans*-caryophyllene, *δ*-cadinene, α-copaene, caryophyllene oxide, germacrene-D, and α-humulene [30]. It is not known, however, which compound is primarily responsible for the allelopathy of *C. odorata*, and it is possible that the effect may be brought about by the joint action of several compounds. Compatible with this speculation, Inderjit et al. [31] mentioned that some allelochemicals might work synergistically or interact between compounds to form a new compound with inhibitory effects. Chotsaeng et al. [32] reported that binary mixtures of R-(+)-limonene:xanthoxyline, vanillin:xanthoxyline, and xanthoxyline:linoleic acid exhibited stronger allelopathic activities on germination and seedling growth in *A. viridis* than single compounds. The hexane fraction was further formulated to suspension concentrate.

The formulation, which contained a crude hexane fraction of *C. odorata*, named as CH product, had greater early-post emergence than pre- and post-emergence characteristics (Figure 2, Figure 3, Figure 4 and Figure 5). The CH product caused leaf wilting at 6 h after application and death between 4 and 7 days after application. The physiological mechanisms causing weed death were determined by the pigment’s contents and the TBARS content in *E. crus-galli* leaves. Our results showed that the contents of chlorophyll a, b, and carotenoid were reduced with increasing times of application, indicating that the CH product interfered with photosynthetic metabolism (Figure 6). This observation was similar to many other reports regarding essential oils and the reduction of the chlorophyll and carotenoid contents in plants, such as *Tagetes erecta* [33], *Cupressus macrocarpa* Hartweg, *Murraya koenigii* (L.) Spreng, *Plectranthus amboinicus* (L.) Spreng, *Persicaria odorata* (L.) Sojak, and *Pelargonium radula* (Cav.) [34]. Chlorophyll a, b, and carotenoid are specific major pigments in plants detected in the chloroplasts that capture light energy [35]. However, whether the reduction in the chlorophyll content is due to a decrease in de novo biosynthesis of chlorophylls or its enhanced degradation is poorly understood.

The sample leaves of *E. crus-galli* were found to contain TBARs content when increasing in the times after CH application, suggesting that lipid peroxidation occurred (Figure 7). Lipid peroxidation refers to the oxidative degradation of lipids. The TBARS assay is the test most used for malondialdehyde (MDA) content, the end-product of lipid peroxidation [36]. Our result agreed with others’ results that showed that some essential oils and terpenes could induce lipid peroxidation on bioassay species, such as *Cymbopogon citratus* [37], *Tagetes erecta* [33], *Artemisia scoparia* [38], and *Foeniculum vulgare* Mill [39]. Fatty acids and other lipids are components of the membrane structure. The results showed free lipids within the cytoplasm, which may indicate membrane degradation caused by oxidative action. The accumulation of oxidative oxygen species, such as H_2_O_2_, caused increased oxidative stress and led to the disruption of cell metabolic activities [40].

## 4. Materials and Methods

### 4.1. General Preparation

*Chromolaena odorata* was collected from an experimental field at Naresuan University, Phitsanulok, Thailand, during August 2020. Mature whole plants were harvested, cleaned with tap water, and separated into leaf, stem, and root. The samples were dried at 45 °C for 72 h and then ground into powders using an electric grinder. The seeds of *E. crus-galli*, a monocot weed, and *A. viridis*, a dicot weed, were collected from an agricultural field in Phitsanulok province. *Echinochloa crus-galli* seeds used for the experiments were aired under full light for 72 h and incubated at 50 °C for 24 h to break their dormancy.

### 4.2. Different Part Extract Bioassay

Powders from the whole plant, leaf, stem, and root of *C. odorata* were separately extracted with 70% aqueous ethanol (10:100, *w*/*v*) at room temperature for 72 h. Each extract was filtered sequentially through cheesecloth and filter paper (Grade 1, Whatman Inc., Buckinghamshire, UK). The residues were extracted twice more with 70% aqueous ethanol. The three solutions were combined and evaporated using a rotary evaporator to produce the crude extracts. All the extracts were redissolved in 70% aqueous ethanol to compare their phytotoxic effects. An aliquot (5 mL) of each solution at different concentrations (1000, 2000, 4000, and 8000 ppm) was added to a 9-cm Petri dish lined with two germination papers, and the solvent was evaporated in a fume hood. After evaporation, distilled water (5 mL) was added to the germination paper, and 20 seeds of either *A. viridis* or *E. crus-galli* were evenly placed on the paper. Distilled water was used as the control. Four replicate experiments were conducted in a randomized fashion for each concentration of each extract. The dishes were incubated under laboratory conditions at room temperature (25–30 °C). After 7 days of incubation, the germination and the shoot and root lengths of test plants were measured and calculated to the percentage of inhibition and the concentration required for 50% inhibition (IC_50_ value).

### 4.3. Sequential Solvent Extraction of Active Compounds and Bioassay in the Laboratory

A partial separation of active compounds was achieved by sequential solvent extraction, using organic solvents with increasing polarity adapted by Shui [41]. The extraction protocol is shown in Figure 8. The dried leaf material of *C. odorata* was first extracted with ethanol for 24 h. The mixture was filtered through Grade 1 filter paper, and the residue was extracted twice more with ethanol. The three solutions were combined and evaporated at 40 °C using a rotary evaporator to produce the crude ethanol extract, which was then divided into two parts. The original crude ethanol extract (OR) in the first part was used to determine phytotoxic effects. The remaining part of the crude ethanol extract was suspended in 90% aqueous ethanol and partitioned with hexane (HX) to give an HX fraction and a 90% ethanol fraction. The ethanol fraction was concentrated using a rotary evaporator. The concentrate was suspended in distilled water and continuously partitioned with ethyl acetate (ET), giving an aqueous and an ET fraction. The aqueous fraction was separated, and the aqueous fraction was then continuously partitioned with *n*-butanol (BU), giving an aqueous and a BU fraction. The aqueous fraction was discarded. The HX, ET, and BU extracts were evaporated using a rotary evaporator to provide HX, ET, and BU crude extracts. All crude extracts were stored in airtight containers under cool and dark conditions until further study.

The OR, HX, ET, and BU crude extracts were redissolved in the original extraction solvent to compare their phytotoxic effects. An aliquot (5 mL) of each solution at different concentrations (500, 1000, 2000, and 4000 ppm) was added to 9-cm Petri dishes lined with two germination papers and evaporated to dryness in a fume hood. After evaporation, distilled water (5 mL) was added to the germination paper, and 20 seeds of either *A. viridis* or *E. crus-galli* were evenly placed on the paper. Distilled water was used as the control. The experimental design, incubation conditions, and data collection were the same as in the bioassay described above.

### 4.4. Suspension Concentrate Formulation

The HX fraction was selected for formulating the available product because it showed the most activity. Water-based suspension concentrate (SC) formulations were made by mixing crude HX (active ingredient) 30%, NP-40 nonylphenol ethoxylate 3%, propylene glycol 6%, bentonite clay 6%, with the addition of deionized water to 100%, which gave the *C. odorata* crude hexane (CH) 30% of the active ingredient (ai). The product was kept at room temperature (25–30 °C) and protected from light before use.

### 4.5. Herbicidal Characteristic

The herbicidal characteristics of a spray of the CH product (SC formulation) of *C. odorata* applied pre- and post-emergence were evaluated.

A plastic pot (radius 20 cm, height 15 cm) was filled with dried field soil (1.2 kg). The soil characteristics were as follows: pH: 6.53; NO_3_-N (mg 100/g dry soil): 20.0; OM (mg 100 g/dry soil): 2.21; exchangeable K_2_O (mg 100 g/dry soil): 103.42; and available P_2_O_3_ (mg 100 g/dry soil): 7.51. Twenty-five seeds of *E. crus-galli* or *A. viridis* were sowed at a depth of 0.5 cm.

For pre-emergence tests, the pots were watered for 1 day, and different concentrations (2.5, 5, 10, and 20 kg ai/ha) of CH product solution were sprayed onto the soil surface with a flat fan nozzle using a water spray at a rate equivalent to 500 L/ha. The pots were not watered for 24 h after being sprayed. The pots were placed in a glasshouse at temperatures between 28 and 35 °C. An application of distilled water was used as the control. The experiment was repeated 6 times, with the trays planted in a completely randomized design. Weed emergence was counted every 2 days until 21 days after application, and in preparation to calculate the dry weight of the shoots, they were collected and dried in an oven at 80 °C.

To test early post- and post-emergence weed stages, the pots were prepared according to the same method as the pre-emergence pots. Seven days after sowing, the emergent *E. crus-galli* and *A. viridis* seedlings (2 leaf stages) were assigned as early-post emergence, while post-emergence was at 25 days old (6 leaf stages). The treatments were as for the pre-emergence experiment. The CH product solution was sprayed onto the pot with a full cone nozzle using a water spray at a rate equivalent to 500 L/ha at a pressure of 2.5 bar. The pots were not watered for 24 h after being sprayed. The pots were placed in a glasshouse at temperatures between 28 and 35 °C. An application of distilled water was used as the control. The experiment was repeated 6 times, with the trays planted in a completely randomized design. The weed control efficacy was estimated at 1, 3, 7, 10, and 14 days after application, with 0% indicating no control or no injury, while 100% represented complete control. The biomass of the shoots was collected from survival seedlings at 21 days after application, which was dried in an oven at 80 °C.

In another set of experiments, the effects of foliar-applied CH product on the growth and physiological mechanisms of *E. crus-galli* were studied. Ten 14-day-old *E. crus-galli* seedlings were prepared as in the previous experiment. The treatment used in this experiment was 5 kg ai/ha of CH product, which was dissolved in distilled water and sprayed, with only distilled water used as the control. A handheld pressure sprayer with a solid cone nozzle was used to spray at a rate of 500 L/ha. Four replicates per treatment were maintained in a completely randomized design. All treatments, including the distilled water control, were sprayed in the randomized pots. The content of chlorophyll a and b, carotenoid, and lipid peroxidation was determined at 12, 24, 48, and 72 h after being sprayed.

### 4.6. Estimation of Chlorophylls and Carotenoids Content

One-hundred-mg fresh leaf samples from *E. crus-galli* in the treatments were extracted with aqueous 80% acetone using a mortar and pestle, and the suspension was filtered through Grade 1 filter paper. The chlorophyll and carotenoid contents were determined spectrophotometrically using the spectrophotometer (Shimadzu UV-1800, Shimadzu Corporation, Kyoto, Japan) at 3 wavelengths: 663 nm for chlorophyll a, 647 nm for chlorophyll b, and 470 nm for carotenoids. Calculations were completed using Lichtenthaler’s equation [42] and expressed as mg/g 158 dry weight.

### 4.7. Lipid Peroxidation

Lipid peroxidation was considered via the content of thiobarbituric acid reactive substances (TBARs), according to the method of Heath and Packer [43]. Three hundred mg of fresh leaves were homogenized in 3 mL of 20% (*w*/*v*) trichloroacetic acid (TCA) and centrifuged at 10,000× *g* for 20 min. One ml of 20% TCA containing 0.5% (*w*/*v*) TBA and 100 μL 4% butylated hydroxytoluene in ethanol was added to 1 mL-aliquots of the supernatant. The mixture was heated at 95 °C for 30 min and then quickly cooled on ice. The contents were centrifuged at 6000× *g* for 15 min, and the absorbance was measured at 532 nm. The value for non-specific absorption at 600 nm was subtracted. The TBAR concentration was calculated using an extinction coefficient of 155 mM/cm. The results were expressed as n mol/g FW.

### 4.8. Statistical Analysis

Data were analyzed using analysis of variant (ANOVA). Whenever ANOVA indicated significant effects (*p* < 0.05), a pairwise comparison of means by Tukey’s studentized range distribution test was conducted. Tukey’s HSD (honestly significant difference at *p* < 0.05) identified any difference between means of the data. All statistical analyses were performed using SPSS version 21.0. The IC_50_ values of each test plant were analyzed by GraphPad Prism 5.04. (GraphPad Software, Inc., La Jolla, CA, USA).

## 5. Conclusions

*Chromolaena odorata* extract has been shown to have allelopathic activity on both *A. viridis* and *E. crus-galli*. The leaf extract showed the highest activity, followed by extracts derived from the whole plant, the stem, and the root. The crude leaf extract was further separated by solvent partition based on solvent polarities. The HX fraction showed higher activity, and the crude HX was, therefore, selected to test for pre- and post-emergence herbicidal activity. The HX crude extract was formulated to a suspended concentrate (SC), producing the CH product. The CH was applied as a soil and foliar spray, and the CH was shown to have both pre- and post-emergence characteristics; however, early post-emergence showed the greatest herbicidal activity. The CH product may disrupt the process of photosynthesis and induce oxidative stress in *E. crus-galli* seedlings, leading to toxicity activity. Based on observations of the physiological mechanisms, it can be concluded that the CH damaged the plant tissues, especially in the cell walls and cell membranes of the leaf, which caused the disturbance of lipids and electrolytic leakage. Another effect of the penetration of CH in plant cells was the presence of damaged chlorophylls, which caused abnormal photosynthesis. Thus, CH is a candidate for application as a weed control agent in crop production in future field experiments.

## Figures and Tables

**Figure 1 plants-10-01609-f001:**
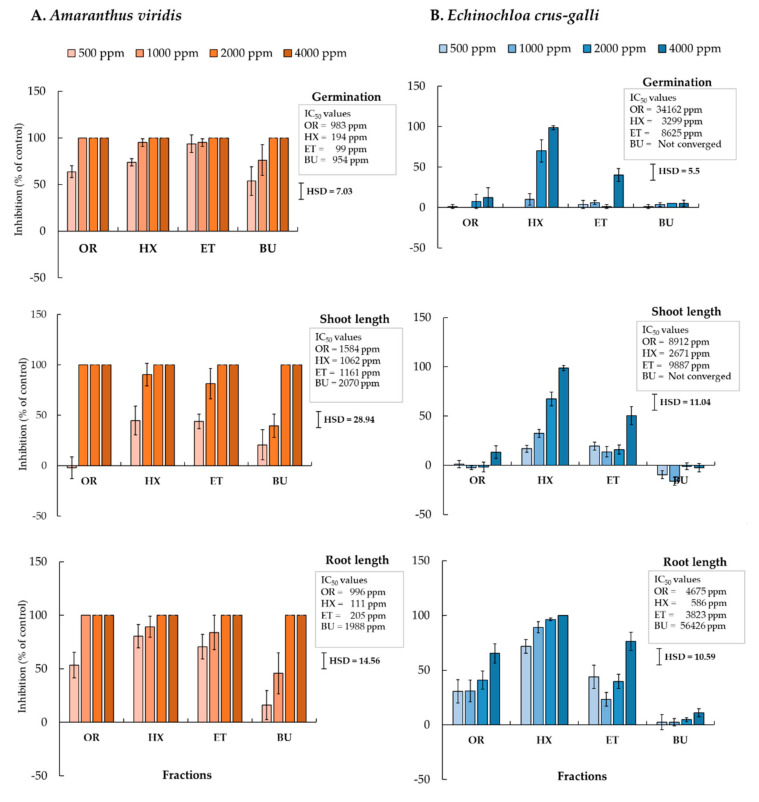
The inhibitory effect of hexane (HX), ethyl acetate (ET), n-butanol (BU), and original ethanol (OR) fractions obtained by *C. odorata* leaves on germination, shoot length, and root length of (**A**) *A. viridis* and (**B**) *E. crus-galli* at 7 days after experiments with Petri-dish test. Means ± SE with four replicates. Tukey’s HSD range test at *p* < 0.05 identified any difference between means of treatments.

**Figure 2 plants-10-01609-f002:**
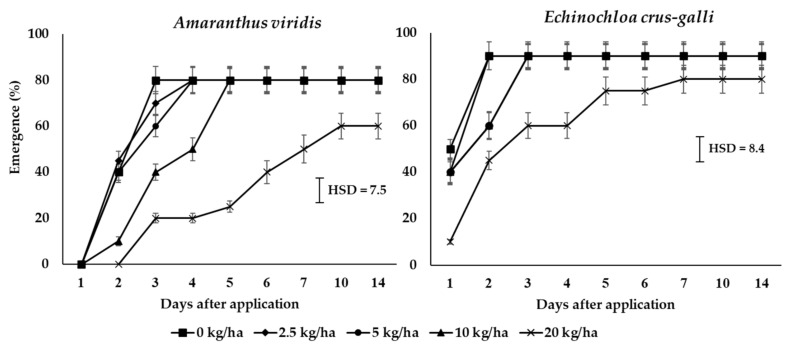
Emergence (%) of *E. crus-galli* and *A. viridis* after pre-emergence spray application of CH product at different days after treatment under pot experiment in glasshouse conditions. Means expressed error bar (± SE) with four replicates. Tukey’s HSD range test at *p* < 0.05 identified any difference between means of treatments.

**Figure 3 plants-10-01609-f003:**
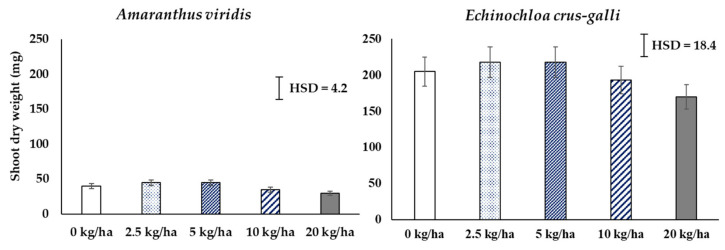
Shoot dry weight of surviving *E. crus-galli* and *A. viridis* seedlings at 21 days after pre-emergence spray application of CH product under pot experiment in glasshouse conditions. Means expressed error bar (± SE) with four replicates. Tukey’s HSD range test at *p* < 0.05 identified any difference between means of treatments.

**Figure 4 plants-10-01609-f004:**
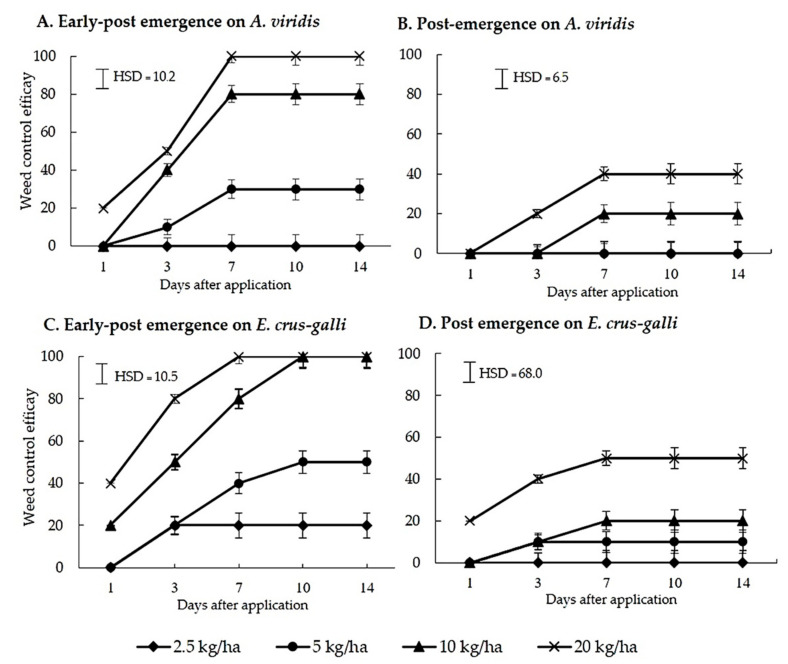
Visible weed control efficacy against *A. viridis* and *E. crus-galli* at various days after early post- and post-emergence spray application of CH product under pot experiment in glasshouse conditions. Means expressed error bar (± SE) with four replicates. Tukey’s HSD range test at *p* < 0.05 identified any difference between means of treatments.

**Figure 5 plants-10-01609-f005:**
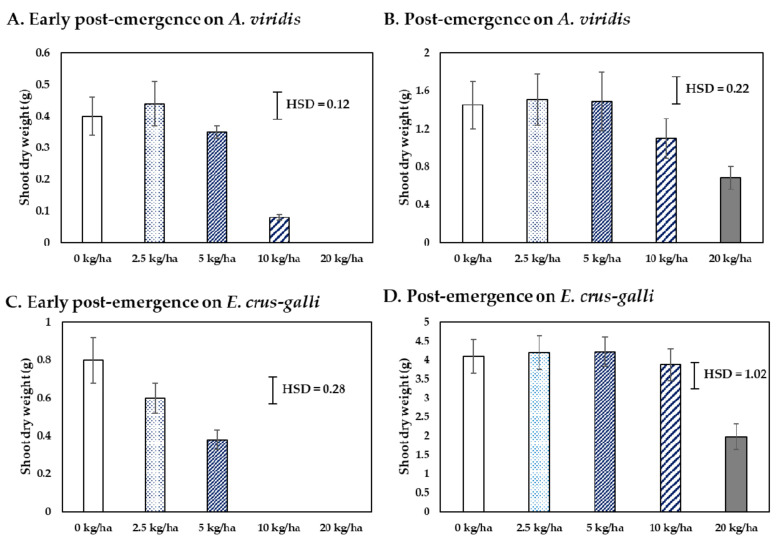
Shoot dry weight of surviving *A. viridis* and *E. crus-galli* seedlings at 21 days after the early post- and post-emergence spray application of CH product under pot experiment in glasshouse conditions. Means expressed error bar (± SE) with four replicates. Tukey’s HSD range test at *p* < 0.05 identified any difference between means of treatments.

**Figure 6 plants-10-01609-f006:**
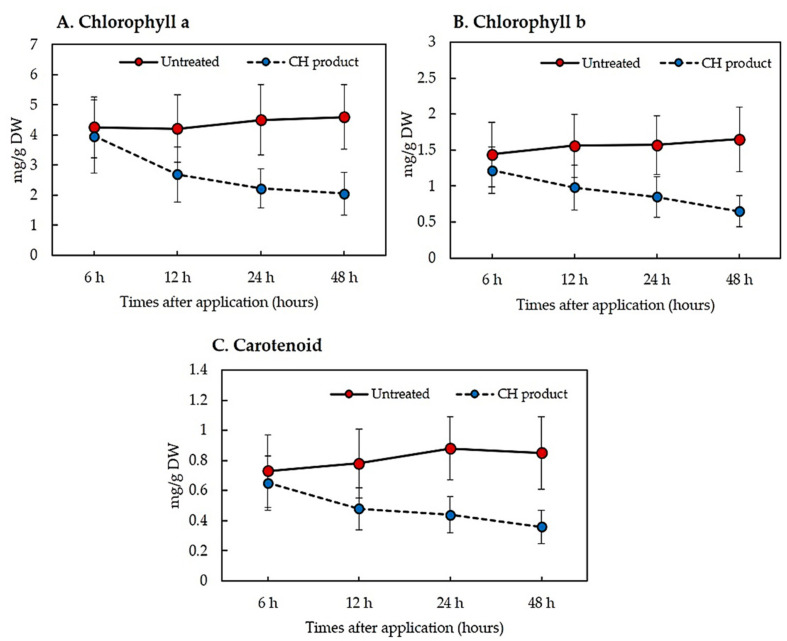
The amounts of chlorophyll a (**A**), chlorophyll b (**B**), and carotenoid (**C**) contents in *E. crus-galli* leaf at increasing times after post-emergence spray application of CH product under pot experiment in glasshouse conditions. Means expressed error bar (± SE) with four replicates.

**Figure 7 plants-10-01609-f007:**
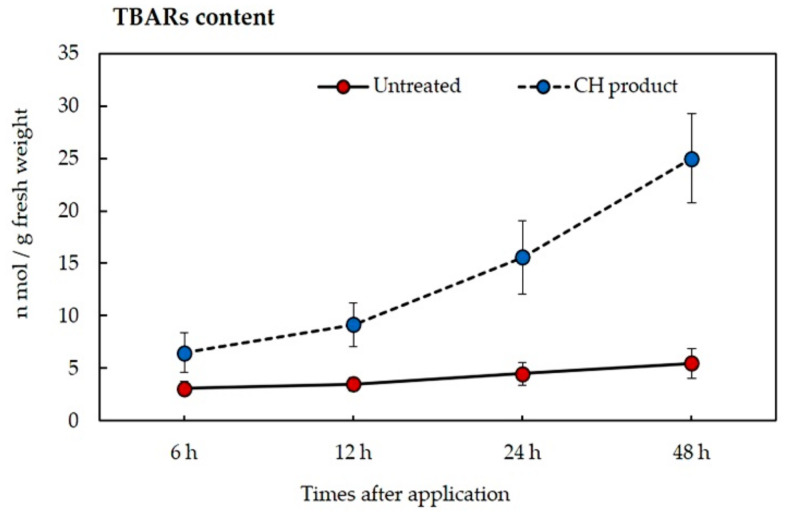
The amounts of TBARs in *E. crus-galli* at the increasing times after post-emergence application of CH product under pot experiment in glasshouse conditions. Means expressed error bar (± SE) with four replicates.

**Figure 8 plants-10-01609-f008:**
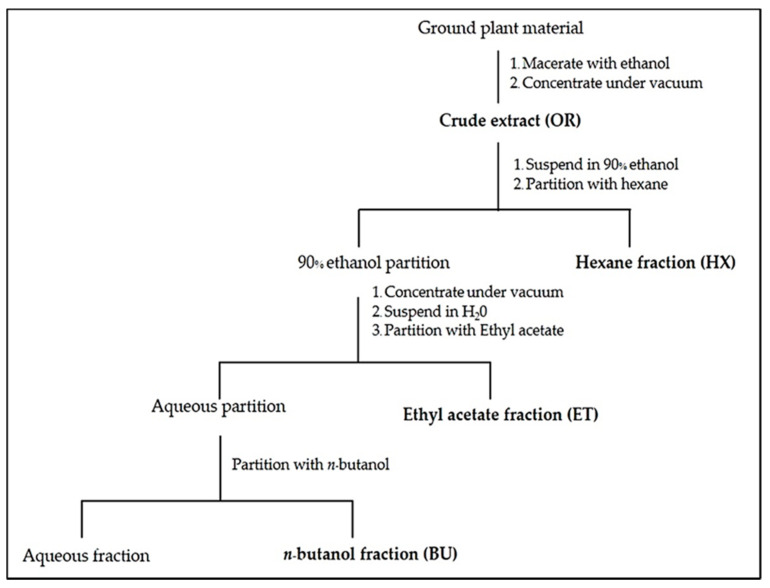
The prepared extracts representing a range of polarities.

**Table 1 plants-10-01609-t001:** The inhibitory effect of crude ethanol extract obtained by different parts from *Chromolaena odorata* on germination, shoot length, and root length of *Echinochloa crus-galli* and *Amaranthus viridis* at 7 days after experiments with Petri-dish test.

Concentration	*Amaranthus viridis*			*Echinochloa crus-galli*		
	% Inhibition			% Inhibition		
	Germination	Shoot Length	Root Length	Germination	Shoot Length	Root Length
Shoot extract						
1000 ppm	33.7 ± 8.5	−74.3 ± 20.2	22.7 ± 4.9	2.5 ± 2.8	−2.7 ± 3.0	−8.1 ± 1.2
2000 ppm	51.2 ± 9.4	−18.0 ± 26.0	18.7 ± 6.5	5.0 ± 0.0	6.1 ± 2.0	9.8 ± 2.1
4000 ppm	93.7 ± 2.5	31.7 ± 12.5	21.9 ±10.1	5.0 ± 0.0	29.6 ± 7.3	9.9 ± 1.7
8000 ppm	100 ± 0.0	100 ± 0.0	100 ± 0.0	3.7 ± 2.5	70.3 ± 6.7	72.5 ± 3.1
IC_50_ (ppm)	1610	4177	4607	Not converged	5671	6332
Leaf extract						
1000 ppm	92.5 ± 2.8	68.7 ± 16.5	15.2 ± 8.1	6.2 ± 2.5	43.6 ± 9.4	47.7 ± 7.7
2000 ppm	100 ± 0.0	100 ± 0.0	100 ± 0.0	2.5 ± 2.8	53.1 ± 5.8	69.8 ± 4.5
4000 ppm	100 ± 0.0	100 ± 0.0	100 ± 0.0	91.2 ± 4.7	90.5 ± 5.2	98.1 ± 1.5
8000 ppm	100 ± 0.0	100 ± 0.0	100 ± 0.0	92.5 ± 5.0	92.3 ± 5.0	99.6 ± 0.6
IC_50_ (ppm)	862.3	975.5	1049	3047	1367	1101
Root extract						
1000 ppm	30.0 ± 7.0	−32.6 ± 22.5	11.1 ± 10.3	2.5 ± 2.8	22.3 ± 6.2	2.1 ± 4.4
2000 ppm	52.5 ± 6.4	3.0 ± 0.5	12.5 ± 12.3	1.2 ± 0.5	28.3 ± 5.9	7.5 ± 4.5
4000 ppm	100 ± 0.0	100 ± 0.0	100 ± 0.0	3.7 ± 1.4	29.7 ± 7.3	18.5 ± 5.1
8000 ppm	100 ± 0.0	100 ± 0.0	100 ± 0.0	5.0 ± 2.4	46.5 ± 6.7	26.9 ± 1.2
IC_50_ (ppm)	1633	2385	2366	Not converged	13,126	19,473
Whole plant extract						
1000 ppm	33.7 ± 6.2	−8.2 ± 14.0	34.7 ± 11.3	0.0 ± 0.0	26.4 ± 4.1	15.3 ± 4.1
2000 ppm	81.2 ± 4.7	60 ± 17.5	35.9 ± 12.3	1.2 ± 0.5	39.6 ± 6.2	46.5 ± 4.5
4000 ppm	100 ± 0.0	90.4 ± 19.1	100 ± 0.0	6.2 ± 0.5	39.6 ± 6.3	35.7 ± 7.1
8000 ppm	100 ± 0.0	100 ± 0.0	100 ± 0.0	1.2 ± 0.5	52.1 ± 6.5	68.4 ± 5.3
IC_50_ (ppm)	1241	1965	1856	Not converged	7204	4310
HSD	15.4	28.5	22.2	17.8	21.6	24.5

Means ± SE with four replicates. Tukey’s HSD range test at *p* < 0.05 identified any difference between means of treatments.

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
