# Peer review of "Allelopathic and Herbicidal Effects of Crude Extract from Chromolaena odorata (L.) R.M.King and H.Rob. on Echinochloa crus-galli and Amaranthus viridis"

_plants, 2021, doi:10.3390/plants10081609_

Round 1

Reviewer 1 Report

Manuscript is perfectly arranged and I am not against to publish it. But authors need to explain why they used presented methods. Especially to explain the control variants.

Why it was not used something as reference? It means some well-known herbicide?

Distillation water was used as a control? Why not to summit all extractions to some other plant and this use as a control?

Untreated means not treated with anything, but comparison treated-non treated did not exclude the influence of solvent.

Authors need to explain these methodological questions.

Incubation of seeds at 50°C need some more explanation. Did they survive? Maybe it was tested by another scientists, but this needs discussion. In normal case plant material is not surviving in such high temperatures.

Table 1

I am missing 0,000. What was the chemical composition of each extract? Why it was not used the extract from any other plant as a control variant?

Figure 1

I am missing reference variant (zero variant, commercial herbicide variant)

Figures 2 -6

There is only the test of CH product and zero application. Where is reference variant?

Author Response

Why it was not used something as reference? It means some well-known herbicide?

            - Our research intended to understand the efficacy and characteristics of crude extract from Chromolaena odorata. Thus, we did not compare with other herbicides. However, we intend to compare with commercial in a crop production on next experiment.

Distillation water was used as a control? Why not to summit all extractions to some other plant and this use as a control?

            - The distillation water was the negative control because it is not any compounds content it.

Untreated means not treated with anything, but comparison treated-non treated did not exclude the influence of solvent.

            - We observed on preliminary test about the effect of inert ingredient (solvent, surfactant, and clay powder). It was showed no-effect against bioassay species.

Authors need to explain these methodological questions.

Incubation of seeds at 50°C need some more explanation. Did they survive? Maybe it was tested by another scientists, but this needs discussion. In normal case plant material is not surviving in such high temperatures.

- For Echinochloa crus-galli seeds. It was a standard method to break seed dormancy. The result showed that the germination of control (distilled water) was about 80 – 100%.

Table 1

I am missing 0,000. What was the chemical composition of each extract? Why it was not used the extract from any other plant as a control variant?

- The other plant extracts also contain any substances which inhibited or stimulated against tested species.

Figure 1

I am missing reference variant (zero variant, commercial herbicide variant)

Figures 2 -6

There is only the test of CH product and zero application. Where is reference variant?

               - As we explained previous question. Our research intended to understand the efficacy and characteristics of crude extract from Chromolaena odorata. Thus, we did not compare with other herbicides.

Reviewer 2 Report

Allelopathic and Herbicidal Effects of Crude Extract from Chromolaena odorata (L.) King & Robinson on Echinochloa crusgalli and Amaranthus viridis

The present study was aimed at investigating the allelopathic effects of crude extracts from different parts of an invasive weed Chromolaena odorata on the germination and growth of Echinochloa crus-galli and Amaranthus viridis.   The paper reports the results of an extensive set of experiments which examine multiple components of the system.   Overall, the results clearly demonstrate that extracts of C. odorata have inhibitory effects against the two test plant species at multiple levels, and thus there is potential for these compounds to be developed as botanically-derived herbicides. 

In general, the writing is OK, but there are occasions where the text does not make sense or details have been omitted (eg line 51).  In the results, the titles to figures and tables need much more detail to explain better what they are showing.

In the Introduction , there is no indication as to why A. viridis and E. crusgalli were chosen as test plants

Also, I am uncertain as to whether the authors are actually examining allelopathy, or just bioactive compounds or phytotoxic effects in the various extracts.  For example, many Piper plants produce insect repellent extracts, but this is not usually considered allelopathy?  Botanically-derived herbicides do not necessarily need to involve allelopathic comunds?

The authors need to be careful with latin names;

- do not start sentence with abbreviated name (eg line 259)

- make sure abbreviated genus names cannot be confused  eg line 97 A. viridis, but the last genus name used beginning with A was Ageratum on line 82.  Use full genus name if unsure.

Also use full genus name in figure and table titles.

Detailed comments

Abstract

Ln 25 odorata

Intro

Ln 51 needs correcting

Ln 56 use full genus name for E. crus-gallis as it is first mention in main text?

Ln 67 insoluble in water

Ln 84 extracts not italics

Say why you used the two test plants – the method section comes too late

Methods

Ln 259 & Ln 264 & ln 393 & ln 100 full genus name

Ln 265 Is 50oC correct – does this not cook the seeds?!

Ln 274  was there both a 0 ppm control treatment with just solvent?   And a ‘blank’ with just water on the germination papers?

Ln 386 variance

Results

Table 1 put full species names in title. 

And say this was in Petri dish trials, and at what duration the results were obtained.  

Is the control treatment included in the ANOVA and the Tukey tests?

Maybe replace the letter codes to show which treatments are significantly different from the control, not each other (ie Dunnets test, not Tukey test).

As some results are all 100% with no variation within treatments, is ANOVA actually valid here?

The negative values at low doses suggest a significant benefit is occurring ? 
Why is this?  Do small doses actually promote plant growth?  

All figure legends need much more detail.

Fig 1.  Y axis from -50 to 150 seems odd when dealing with % inhibition.

As above, could you just indicate which treatments are different from the control, rather than use the letter codes.  It is slightly confusing because the control dose is not present in the figure.

Fig 2.  What does HSD mean – not in title?

Indicate that these plants are grown in pots or petri dishes? 

What is the sample size?

Are you indicating means?  With se’s or 95% Cis?

Y axis lable    Emergence (%)

Fig 3.  Details missing as for Fig 2.

Y axis lable    Shoot dwt (mg)

Figs 3 & 5 are bar graphs so why not use letter codes as in previous bar graphs rather than HSD bar?

Try and keep style the same throughout paper.

Fig 6 & 7.  Needs sorting as for Fig 2.  More information required.

After application of what???!!

It’s not continuous time?

Ln 173 needs correcting

Discussion

First two sentences need rewriting so as to make sense.

Start with summarizing the results?

First paragraph too long – split in to comprehensible chunks.

Seems odd having the conclusions separate from the Discussion.

Author Response

Dear reviewer 2

I attached the response file and the edited manuscript.

Best regards,

Authors

Round 2

Reviewer 1 Report

Manuscript is perfectly arranged and I am not against to publish it. But authors need to explain why they used presented methods. Especially to explain the control variants.

Why it was not used something as reference? It means some well-known herbicide?

Distillation water was used as a control? Why not to summit all extractions to some other plant and this use as a control?

Untreated means not treated with anything, but comparison treated-non treated did not exclude the influence of solvent.

Authors need to explain these methodological questions.

Incubation of seeds at 50°C need some more explanation. Did they survive? Maybe it was tested by another scientists, but this needs discussion. In normal case plant material is not surviving in such high temperatures.

Table 1

I am missing 0,000. What was the chemical composition of each extract? Why it was not used the extract from any other plant as a control variant?

Figure 1

I am missing reference variant (zero variant, commercial herbicide variant)

Figures 2 -6

There is only the test of CH product and zero application. Where is reference variant?

Author Response

Dear Reviewer 1

Thank you very much for your suggestions. I attached the file of response comments to you. Please see the attachment and consider. 

Best Regards,

Corresponding author.
